When does self-report of pain occur?: A study of older adults

Rodríguez Iyubanit iyubanit.rodriguezramirez@ucr.ac.cr 1
Cajamarca Gabriela 2
Herskovic Valeria 2
1 Department of Engineering, Universidad de Costa Rica , Alajuela , Alajuela , Costa Rica
2 Department of Computer Science, Pontificia Universidad Católica de Chile , Santiago , Región Metropolitana , Chile
Shang Yilun
Electronic publication date: 2022 Jul 19
Publication date: 2022
Volume: 10
Electronic Location ID: e13716
Received 2022 Feb 10; Accepted 2022 Jun 21
Copyright: ©2022 Rodríguez et al.
Copyright year: 2022
Copyright holder: Rodríguez et al.
License: This is an open access article distributed under the terms of the Creative Commons Attribution License, which permits unrestricted use, distribution, reproduction and adaptation in any medium and for any purpose provided that it is properly attributed. For attribution, the original author(s), title, publication source (PeerJ) and either DOI or URL of the article must be cited.
License URL: https://creativecommons.org/licenses/by/4.0/

Keywords: Older adults, Self-report, Tangible interfaces, Chronic pain

Funding: CONICYT-PCHA/Doctorado Nacional/2014-63140077 CONICIT and MICIT Costa Rica PhD scholarship grant Universidad de Costa Rica ANID Fondecyt 1211210 ANID-PFCHA/Doctorado Nacional/2018-21180784 This project was supported by CONICYT-PCHA/Doctorado Nacional/2014-63140077, CONICIT and MICIT Costa Rica PhD scholarship grant, Universidad de Costa Rica, ANID Fondecyt 1211210, and ANID-PFCHA/Doctorado Nacional/2018-21180784. There was no additional external funding received for this study. The funders had no role in study design, data collection and analysis, decision to publish, or preparation of the manuscript.

==============================
Technologies for self-care can drive participatory health and promote independence of older adults. One self-care activity is regularly measuring and registering personal health indicators (self-reporting). Older adults may benefit from this practice, as they are more likely to have chronic health issues and have specific self-monitoring needs. However, self-reporting technologies are usually not designed specifically for them. Pain is usually measured using patient reports compiled during medical appointments, although this process may be affected by memory bias and under reporting of fluctuating pain. To address these issues, we introduced a simple tangible interface to self-report pain levels and conducted a three-hour evaluation with 24 older adults. The goal of this study was to identify whether specific activities, activity levels or pain levels trigger older adults to self-report their pain level, besides to understand how older adults would use such a device. Within the limited time frame of the experiment, the majority of our participants chose to report pain when they felt it most, not reporting lower levels of pain. No evidence was found to suggest a relationship between the reporting of pain and the activity (or activity level). Several design insights intended to improve the design of technologies are provided.

Introduction

The population of older adults is growing, as is the cost of their medical care. Technology may help reduce the needs for medical intervention, easing the pressure on medical systems, so it is becoming an increasingly important element in supporting self-care (Chernbumroong, Atkins & Yu, 2010). Technologies for self-care are beneficial because they can drive participatory health and promote independence (Chernbumroong et al., 2013), enabling older adults to detect changes in their health (Caldeira, Bietz & Chen, 2016). One self-care activity is self-monitoring (or self-reporting), in which the person regularly measures and registers a particular health indicator (e.g., blood pressure) (Caldeira et al., 2017). However, technologies that allow older adults to register, or monitor, their health indicators, are usually not designed for their specific needs (Davidson & Jensen, 2013).

Pain is a common occurrence among older adults (Patel et al., 2013) and because it is subjective, its self-report is a crucial part of medical visits (Adams et al., 2017). During medical appointments, patients are asked to describe how much pain they have experienced over a certain period of time. However, this process can be inaccurate, e.g., due to the “peak-end phenomenon”, in which patients typically only recall their worst or most recent levels of pain (Fillingim et al., 2016). This phenomenon might even more pronounced in people with cognitive impairment, and as this information is frequently recorded on paper, it can be lost or difficult to analyze in detail (Spyridonis et al., 2014) to understand patterns over time. Furthermore, pain can vary according to time of day and modality (Aviram, Shochat & Pud, 2015), e.g., depending on the activities carried out by the patient.

Diaries allow patients to record their pain during their daily activities, gathering a more accurate and complete overview of their pain levels. Diaries have been found to reduce pain and improve mood in patients with chronic non-cancer pain (Chareonpol et al., 2019) and help cancer patients and their caregivers manage pain in a better way (Valenta et al., 2021). Technological implementations of diaries (e.g., digital diaries (Lazaridou et al., 2018) and mobile applications (Spyridonis et al., 2014; Bird et al., 2016)) have been created to ease pain reporting. These tools can produce benefits at the personal level, e.g., by improving quality of life (Adams et al., 2017). However, a recent review of pain-reporting applications with digital manikins found that they still need further functionalities (e.g., providing useful summary visualizations) as well as to include users in their development (Ali et al., 2021).

It is important to understand the perceptions of older adults regarding technology in order to design tools in an age-friendly way (Vaportzis, Giatsi Clausen & Gow, 2017). When designing technology, it is also important to take into account users’ possible physical and cognitive limitations (Nedopil C., 2013)—e.g., the use of a touchscreen can prove particularly challenging for older adults (Motti, Vigouroux & Gorce, 2013; Sciarretta et al., 2015). Although age does not determine the level of digital skills—rather, interest in innovation is a more determinant factor in technological acceptance (Nedopil C., 2013) - the design process must also take into consideration that some older adults are lacking in digital skills (Padilla-Gngora et al., 2017; Roupa et al., 2010), and have trouble e.g., inputting information into digital interfaces (Doyle et al., 2014). Since unfamiliar technologies are associated with user anxiety (Iancu & Iancu, 2017), and lack of confidence may cause rejection of technology (Nedopil C., 2013), it is important to design technologies that do not frustrate or intimidate users. Some researchers suggest that older adults may prefer to use tangible user interfaces (TUIs) (Nilsson, Johansson & Håkansson, 2003; Wolfgang, 2011) that allow for a physical interaction with digital information, since they can interact with such devices in a way that is familiar to them. Tangible devices have been found to be easier to use for older adults to report pain in hospital settings (Price et al., 2018), and as they can be used discreetly, they may be appropriate for self-reporting (van Berkel et al., 2022).

Several studies that use interfaces for self-reporting pain ask the user to report on a regular basis, or send them reminder notifications to do so, which means the user does not have to decide when to report their pain. Some devices use predefined times (Trudeau et al., 2015), ask periodically after a certain number of hours (Price et al., 2018), or a certain number of times per day (Adams et al., 2017). One study was implemented with three settings (home environment, hospital environment and a combined hospital-and-home environment), asking some users to report at a predefined time of day and others to report when necessary (Alakarppa, Riekki & Koukkula, 2009). However, the time at which pain was reported (and any events that triggered reporting) was not analyzed. One recent prototype interface allows users to record pain when they are experiencing an event through fist clenching gestures (Ramprasad et al., 2021), but this interface has not yet been tested with users.

This study aims to provide understanding about how older adults use a simple tangible interface to self-report pain. Specifically, we focus on the usability of such a device, as well as the moment in which the older adult reports pain, to determine if activities or pain levels trigger self-reporting. Understanding what guides people to make decisions about pain is essential to understanding the process of pain reporting (Dildine, Necka & Atlas, 2020). Therefore, studying the context of the self-report of pai can provide us with greater knowledge of the factors that trigger it, which could provide healthcare workers information on how to improve a patient’s treatment; as well as providing design guidelines that may aid the implementation of these types of devices. To explore these issues, we designed a simple tangible interface called TeRa (based on an interface used in a previous study (Rodríguez et al., 2017a), called RepWear), which allows users to self-report pain levels, and asked older adults to try it for a short period of time (3 h). The research questions guiding this research are the following ones: (1) What event triggers older adults who experience chronic pain and have low digital skills to report their pain? (2) What is older adults’ perception of a tangible device for reporting pain?

The rest of the article is organized as follows. ‘Materials and Methods’ describes the TeRa prototype and its evaluation. ‘Results’ presents the results, while ‘Discussion’ presents the discussion, including limitations of this work. Finally, ‘Conclusions’ describes our conclusions and possible avenues of future work.

Materials and Methods

This section describes the design and implementation of TeRa, a simple tangible device for users to record their pain levels. Then, we describe how the device was used to understand which activities or events may trigger participants to register their pain levels. For this, we decided to monitor contextual information using a context model for adults with pain (Rodríguez et al., 2017b) as a point of reference, i.e., we selected three contextual variables: emotional state, patient activity and heart rate. These variables were measured with other devices or techniques, not through the TeRa device, which only records pain levels. Physical activity and heart rate were captured in real time, while emotional state was obtained through a questionnaire, applied at the beginning of the study. Participants in the study were asked to use TeRa to register their pain whenever they wanted while they carried out their usual activities for three hours.

TeRa: design and implementation

We implemented a tangible interface for self-reporting pain, called TeRa. The device allows users to record their pain level by using a numeric rating scale (NRS), which requires patients to assign a number from 0 to 10 to their pain (in which 0 = no pain, and 10 = the worst pain imaginable). The NRS is the gold standard for pain reporting (Suzuki, 2017) and has been used widely with older adult populations (Flaherty, 2008). The device is based on interface design principles for older adults. We used two main guiding principles to create the design: (1) the device must be simple and easy to use, without excessive instructions (Pyae et al., 2016); and (2) the device should use a concept that is familiar to older adults (Abdul et al., 2012). For these reasons, and based on previous studies (Rodríguez et al., 2017a; Rodríguez et al., 2016), we developed a prototype with only two push buttons, each of which is similar to two physical buttons on a radio (play, record), to achieve a simple interface that requires minimal cognitive effort. The interface has large buttons, enabling an easy tactile interaction (Iancu & Iancu, 2017; Iglesias, Gomez de Segura & Iturburu, 2009) (see Fig. 1) and clear feedback through sound effects (Rodríguez et al., 2018), such as the clicking sound when the button is pressed. We wanted the tangible interface to be small in size, in order to allow it to be transported during the daily routine. Other TUI such as Painpad and Keppi have been proposed for similar purposes. Painpad has been tested in clinical settings, but the size makes it difficult to move (Price et al., 2018); while Keppi version one and two have a suitable size, but they only have four levels of pain (Adams et al., 2018), so it is not compatible with healthcare workers’ practices in our case.

The interface was implemented using LilyPad Arduino, which is a microcontroller board for wearable devices and electronic textiles. It can be sewn to fabric, power supplies and sensors with conductive thread (Arduino, 2022). Specifically, the materials we used are a Lilypad main board (based on the ATmega168V), green and red tactile buttons, a Grove-4-Digit Display module (12-pin), a real-time clock(integrated DS1307), a microSD socket to store the information, and a 1Ah battery.

With TeRa the pain level is initially at five, and the user can press the green button to rotate through pain levels, up to 10 and starting again at 0. When the user locates the intensity of pain that they wish to record, they press the red button (see Fig. 2), thereby storing the intensity of pain and the timestamp.

Figure 1 TeRa: a tangible device to self-report pain.

Figure 2 Participant reporting pain by advancing from six to eight on the pain scale and storing the desired intensity level.

Study context: participants and recruitment

The inclusion criteria for participants were the following: (1) they had to be 60 years old or older, (2) without cognitive problems and (3) experiencing chronic pain. Our final sample of participants included 24 older adults (women = 11, men = 13), between the ages of 60 and 87 (M = 69, SD = 8.14). Eleven participants lived in a retirement home and 13 in their own home; 16 were receiving varying degrees of treatment for pain (medication or physical therapy) during the time of the study; while the remaining participants were receiving no type of treatment during the time of the study, despite experiencing pain (Table 1).

Table 1 Description of study participants.

P	Age	Gender	Educational level	Digital skills	Home	Anxiety	Depression	Pain area	Pain severity	Treatment	
P1	77	F	School	None	Own	Y	Y	Spine, hand, feet	6	Pills, yoga	
P2	60	F	School	Low	Own	N	N	Knees	2.5	Pills	
P3	61	F	Technical	Above basic	Own	N	N	Sciatica	6.7	Pills	
P4	63	M	High school	Basic	Own	Y	Y	Sciatica, shoulder	4.7	Pills	
P5	77	M	School	None	Nursing	N	Y	Tailbone	7.7	None	
P6	73	M	High school	None	Nursing	N	N	Sciatica, knees	0	Physical therapy	
P7	73	M	School	None	Nursing	N	N	Sciatica	0	None	
P8	73	M	School	None	Nursing	Y	Y	Sciatica, shoulder	5.5	None	
P9	65	M	School	None	Nursing	Y	Y	Leg	7.5	Physical therapy	
P10	73	M	None	None	Nursing	Y	Y	Head, belly, leg	4.7	Pills, therapy	
P11	67	M	School	None	Nursing	N	N	Leg	2.7	Pills	
P12	77	M	High school	Basic	Nursing	N	Y	Head, hand, knee	5	Pills, therapy	
P13	66	M	School	Low	Own	N	N	Shoulder	2.7	Pills	
P14	60	F	School	Low	Own	N	N	knee, neck	3	Pills	
P15	76	F	School	None	Own	Y	Y	knee, spine	5	Pills	
P16	60	F	School	Low	Own	Y	Y	knee, shoulder	2.7	Pills	
P17	60	F	School	Basic	Own	N	N	Arm	0	Physical therapy	
P18	65	F	School	None	Own	N	N	Shoulder, wrist	3	Pills, therapy	
P19	60	M	School	None	Own	N	Y	Arm	4.2	Pills	
P20	60	F	School	Low	Own	N	N	Neck, wrist	3	None	
P21	68	M	School	None	Own	N	Y	Waist	0,7	None	
P22	87	M	High school	None	Nursing	N	N	Leg, eye	4.5	None	
P23	72	F	High school	None	Nursing	N	N	Knees	2.7	Pills	
P24	85	F	High school	Low	Nursing	Y	N	Shoulder, back, knee	2.5	Pills	

We recruited participants through two techniques. First, we used snowball sampling in the following way. We contacted a small number of initial participants (seeds), who had participated in previous studies at the university and who met the inclusion criteria. If they agreed to participate in this study, after their participation we asked them to recommend contacts who fit the research criteria and who could potentially participate as well. Second, we recruited participants from nursing homes. To do so, we contacted nursing homes near the university campus and asked whether they were interested in participating. If they expressed interest, we sent them information about the study, e.g., ethics approval and consent forms. If the nursing home accepted, they gave us a list of possible participants who fit the inclusion criteria. Then, the potential participants were asked if they were willing to participate and, if so, we agreed on a schedule to carry out the study.

Data collection: materials

To collect the information from the participants, several instruments were used. Each instrument is described below.

1. Evaluations of the technology:

• System usability scale. The system usability scale (SUS) is used to evaluate the usability of a product or system (Brooke, 1996). The SUS score can range from 0 to 100, and generally, scores below 60 are considered “not acceptable”, while scores over 80 are considered “acceptable” (Tullis & Albert, 2008).

• AttrakDiff questionnaire. AttrakDiff is an online survey to assess users’ feelings about an interactive product. Answers are provided based on a scale of −3 to 3 (0 = neutral). The questionnaire has four dimensions: pragmatic quality, hedonic quality-identity (HQ-I), hedonic quality-stimulation(HQ-S) and attraction (GMBH, 2017; Isleifsdottir & Larusdottir, 2008).

2. Participant characteristics:

• DIGCOMP. DIGCOMP is a digital skills questionnaire used to measure a person’s level of digital competence, classifying users into one of four possible levels: none, low, basic and above basic (Ferrari, 2012).

• Brief Pain Inventory (BPI). BPI is a questionnaire that consists of nine elements, aimed to assess pain levels. It includes questions about pain during the day, pain intensity, pain treatment and pain interference (Poqueta & Lin, 2016).

• Goldberg Anxiety and Depression Scale (GADS). GADS is a questionnaire that can be used to assess the emotional state of a person. It consists of two subscales: one for anxiety and one for depression (Goldberg et al., 1988).

3. Semi-structured interview data. After completing the experience, we briefly interviewed each participant. To conduct the interview, the authors created a question guide, based on our research questions, to obtain data from the participants about their experience using TeRa and when they chose to self-report their pain. The questions are shown in Table 2.

Ethical considerations

The research protocol was approved by the university ethics committee (Ref. 15-339). Each participant received oral and written information about the aim of the research, and written consent to the overall study was subsequently provided. Participants were informed that their involvement was voluntary, that anonymity would be guaranteed, and that they could withdraw from the study at any time.

Table 2 Semi-structured interview question guide.

N	Question	
1	When did you report pain and why?	
2	If you could keep the device, would you continue to use it?
(If yes), How often would you use it?	
3	Did you find the device to be useful?	
4	Would you change anything about the device?	
5	Were you able to correctly express your pain using the numeric scale?	
Age (average)	69 years							
Gender	Female	11	45.8%	Home	Nursing	11	45.8%	
	Male	13	54.2%		Own	13	54.2%	
Digital skills	None	12	50.0%	Educational level	None	1	4.2%	
	Low	8	33.3%		Primary	18	75.0%	
	Basic	3	12.5%		Secondary	4	16.6%	
	Above Basic	1	4.2%		Technical	1	4.2%	
Disabilities	Wheelchair user	4	16.6%	
	Blind	1	4.2%	
	Speech difficulties	2	8.3%	

Experiment

The evaluation was conducted over three months, before the COVID-19 pandemic (WHO, 2020). Each experiment had a duration of between 230 and 245 min per participant (each participant performed the experiment individually). To understand the perceptions of older adults with regard to the moment in which they self-report pain intensity, the following activities were carried out:

First, researchers explained the goal of the study and participants were asked to sign the consent form (10 to 15 min). Then we applied the DIGCOMP, GADS and BPI questionnaires and we captured demographic data, e.g., name, gender, and age (10 to 15 min).

Following this, a researcher explained how the TeRa device works by using an example scenario. This example involved a person who experienced pain and used a device to self-report pain for a predetermined period of time. In their subsequent medical appointment, the physician was able to see the pain scores reported by the patient (10 to 15 min).

Participants were then handed the TeRa device and asked to carry out their normal routine (Fig. 3). During the evaluation, the device was placed inside a waist pouch and participants were asked to self-report pain whenever they wanted to. Participants were also given a Garmin Smart Activity Tracker (vívosmart RH+) wristband, to measure activity and heart rate. During this period of time, the researcher observed from a distance and took notes (3 h).

After they used TeRa, participants completed the SUS and AttrakDiff surveys (15 to 20 min). Finally, we asked participants to answer questions about their experience through a semi-structured interview (see Table 2) (5 to 10 min). One author conducted the interview and transcribed the recordings verbatim. The interviews were conducted in Spanish, the native language of the participants and the researchers. In the remainder of the article, we identify the participants as P1 to P24.

Figure 3 Participants using TeRa while performing daily activities.

Analysis

The quantitative data (tracker data and reported pain levels) were analyzed by using the Chi-square test of independence to determine whether the variables were related or not, with a 95% reliability. The data of five participants was not recorded on the wristband for unknown reasons, therefore this information was not analyzed for these participants. The qualitative (interview) data was analyzed through thematic analysis (Al-Megren, Majrashi & Allwihan, 2021; Rodriguez et al., 2019; Maitreyee & Lindgren, 2021; Cha et al., 2015), which was used to code the analysis of the interview and notes made during the evaluation (Braun & Clarke, 2006). Only one researcher undertook the interview analysis. Initially, eight codes were obtained in the first phase of thematic analysis, but during the theme review phase, one theme was discarded and five themes were reclassified as sub-themes, leaving a total of two themes overall: What event triggers and TeRa use. Data analysis was conducted in Spanish.

Results

The results are organized in four sections. Section 3.1 contains the general data. Sections 3.2 and 3.3 aim to answer each of the research questions (What event triggers older adults who experience chronic pain and have low digital skills to report their pain? and What is older adults’ perception of a tangible device for reporting pain?). Section 3.4 (TeRa usage during the experiment) provides information about how TeRa was used by older adults. Participant quotes, taken directly from the interviews and translated from Spanish to English, are provided to illustrate the discussed concepts.

Participant data

Of the 24 participants, 20 had no or low digital skills. One participant was blind and four were in wheelchairs. It was not possible to conduct an oral interview with two of the wheelchair participants because one participant communicated by means of an alphabet table and another participant had mobility problems and a speech impediment. Both interviews were recorded in writing. Table 1 provides a summary of participant information.

Four participants stated they had never used a pain scale (NRS) to report their pain levels. For example, P15 stated,“they [the doctors] do not ask the question in that way [using a score of 0 to 10], the doctor just asks you if you are in pain, and you answer yes or no”. In cases such as this, it was necessary to explain how the numeric pain scale from 0 to 10 worked, before starting the experiment.

What event triggers older adults who experience chronic pain and have low digital skills to report their pain?

The number of steps and the heart rate during the experiment were obtained from the participants’ wristbands. For each participant, this data, as well as the level of self-reported pain, were analyzed to determine whether there is a relationship between the level of pain felt by the participant and the number of steps or heart rate. This analysis was performed using the chi-square test of independence. No correlation was found between the variables, therefore, they are independent.

We reviewed whether participants living with anxiety and/or depression (emotional state) had higher pain that caused interference in their daily lives (Fig. 4), and although most participants that live with depression and anxiety report a higher incidences of pain that interferes with their day-to-day lives, the trend is not significant. However, pain severity is correlated with pain interference.

Figure 4 Relationship between emotional state, pain level and pain interference, per participant.

Black dots represent participants that have anxiety and depression in their daily life, yellow dots represent participants with either anxiety or depression, and violet dots represent participants with neither anxiety nor depression.

The interviews did provide interesting insights as to when participants reported their pain. Participants felt that they were able to report pain adequately and correctly using the NRS and that their doctors could use this information to better understand their pain levels. Three sub-themes emerged to explain when users reported pain: “When pain is felt”, “When no pain is felt”, and “With an objective”. These are described next.

When pain is felt

The majority of participants reported pain only when they experienced the sensation of pain. For example, certain individuals felt pain while moving or changing position and chose to report their pain levels during these moments. Participants confirmed in interviews that they reported pain when moving after having been sitting in one position for a long period of time or when moving abruptly. It should be noted that they did not report pain when the sensation of pain was absent, i.e., when they felt no pain, they did not report a score of zero.

“Well, when I moved, for example, in the case of my arm that hurts, I felt a lot of pain when moving forcefully, that’s why I reported the pain” (P16).

Other participants self-reported pain during the three-hour period, but without having moved or changed position. For example, one participant with arm ligament problems had received physiotherapy treatment during that week and, as a result of decreased pain in the affected area, was reporting lower levels of pain. Similarly, participant P14 stated clearly that when she felt comfortable (no pain), she did not report pain.

“If I touched it gently... my arm would suddenly hurt a little, which is why I reported it, even though it was only a small pain” (P17).

“I reported when I was in pain, only when I was in pain. I didn’t report feeling pain when I felt good” (P14).

Very few participants claimed to have reported decreasing pain levels, due to having rested or changed activity. Similarly, few participants reported a fluctuation in pain, i.e., when they were in pain and then when they were no longer in pain. Participant 24 exemplifies this fluctuation:

“When I walk a lot or when I’m tense, I feel a small pain in the body, in my upper arm, between a three and a four, and suddenly 0 out of the blue... I reported it when I felt it and when I felt nothing I reported 0” (P24).

When no pain is felt

Some participants, despite not feeling pain during the three hours in which the experiment took place, reported pain (with a value of 0). Participants who did not feel pain used different techniques to report, for example, some reported periodically, while others did so when they thought they should report. The reporting was seen as a task to be completed.

“I think it’s important to understand your own pain levels. I reported every so often to try to see if during the passage of time I had been pain free, and during the whole period I felt no pain” (P18).

With an objective

Participants also reported pain with a specific objective in mind, such as to show the collected information to another person or to themselves.

“To be able to see the pain intensity in one’s actions, step by step” (P4).

What is older adults’ perception of a tangible device for reporting pain?

TeRa had a SUS score of 87.61. Tables 3 and 4 show that users with digital skills ranked none or low and with low educational levels (none or primary) had SUS scores over ninety. However, there was a difference in the outcome of SUS score with respect to older adults living in their homes (94.6) and those who live in a retirement home (79.3).

Table 3 SUS score by digital skills.

Digital skills	Amount	SUS score	
None	12	95.4	
Low	8	90.3	
Basic	3	81.3	
Above basic	1	100	

Table 4 SUS score by educational level.

Educational level	Amount	SUS score	
None	1	90	
Primary	16	94.3	
Secondary	6	72.2	
Technical	1	100	

By analyzing the results obtained from the AttrakDiff questionnaire, which measures TeRa user experience, it was found that this device is within the “desired” product quadrant, having obtained a high pragmatic quality (PQ = 1.76) and a high hedonic quality (HQ = 1.57). This means the device is easy to use and helps users to self-report pain, as well as to feel motivated and stimulated.

Likewise, trust levels were 0.37 for PQ and 0.29 from HQ, which means that although responses were a little bit diverse, they were always within the “desired” quadrant (See Fig. 5). The attractive dimension (ATT) achieved a score of 2.15, which indicates that participants positively evaluated the general appearance of TeRa. The score for “how others perceive the product”, i.e., the hedonic quality identification (HQ-I), was 1.93. The hedonic quality stimulation score (HQ-S) was lower (1.22).

Figure 5 Portfolio with average values of PQ and HQ dimensions and the respective confidence rectangles (results from GMBH, 2017).

Regarding how often they would use TeRa, only one participant (who experienced low levels of pain) said that they would not use the device again. In addition, another participant said they were unsure whether they would use it in the future because measuring pain on a scale was a new concept for them. However, several participants stated that self-reporting pain with the device every day would be the best alternative, “to know how my pain is... I would use it a few times a day” (P23).

TeRa usage during the experiment

This section first provides feedback from the participants about how TeRa could be used in the future. Then, we comment on the actual use observed during the experiment.

How to use TeRa

Some of the participants mentioned that being able to self-report pain could provide them with new information that could help them understand more about their own pain. For example, having data about one’s own levels of pain could enable participants to correlate their pain with the specific activities they have undertaken, as well as allow them to get to know themselves better, understand how their pain evolves, and which parts of the body are most affected.

“It’s useful to be able to see the intensity of the pain one feels day-to-day, or to see the intensity of the activities one usually carries out, as a statistic” (P4: 63).

Certain participants raised the possibility of receiving feedback regarding the information they reported during the experiment, which indicates a desire to become more active in managing their chronic pain. Furthermore, they feel that being in possession of a device such as TeRa would help them feel like they are being listened to by the medical professionals.

“I am aware of being in pain... I would like to have some feedback when my time with the device is over” (P16).

How Tera was used

During the experiment, one researcher observed the participants using TeRa, as mentioned previously, and offered them a waist pouch in which to carry the device during the experiment. However, 11 participants decided not to use the waist pouch. Instead, we identified two other locations that participants used to store TeRa (see blue squares in Fig. 3): (1) eight participants (all of them men) placed the device in their shirt pocket, and (2) three participants placed it on the table closest to them.

Discussion

The self-report of pain depends on several factors and not only the pain itself. People report pain as a psycho-physical consequence, but also for other reasons such as to show the result to other people, or to record the change in pain according to the activity they are doing. However, most participants reported their pain when they felt it, i.e., their trigger was “to feel pain”. This type of reporting of pain, i.e., when feeling the sensation of pain, was used in hospital settings whereby people would report pain to alert the nursing staff and get help to control their pain levels (Alakarppa, Riekki & Koukkula, 2009). In our study, participants were informed at the beginning of the experiment that the collected information could be seen by their doctor in their next medical appointment. This could have encouraged our participants to report pain when they felt it, so that healthcare professionals could subsequently help them to control their pain.

It has been suggested by healthcare professionals that interfaces for self-reporting pain can be used as a form of verifying whether pain worsens or improves (Spyridonis et al., 2014). However, in our case very few participants reported pain when no sensation of pain was experienced or when it diminished. As such, one way of observing that the patient is experiencing pain is to compare the amount of reports submitted. If no report, or very few reports, were submitted, the patient may be improving. Otherwise, additional incentives and notifications would have to be implemented to be able to gather data from when the patient does not feel pain.

The rate of user compliance—whether a patient follows medical advice—with self-reporting is important when planning the long-term adoption of a device such as TeRa. Since our device does not have active notifications, lack of compliance may be caused by forgetfulness rather than deliberate lack of cooperation. Several studies in which users self-report pain have used periodic reminders (e.g., (Price et al., 2018; Adams et al., 2017) to encourage participants to input their information in a timely way, improving compliance. For example, an electronic diary with several features to encourage users to fill out self-reports of pain (e.g., audible alarms, alerts when it was not filled out) achieved 94% compliance (Stone et al., 2003). Notifications, however, may be bothersome due to their loud noises and the fact that they can make recipients feel embarrassed (Price et al., 2018). In a study in which a TUI sent a notification to report pain every two hours, participants would occasionally fail to report due to e.g., interruptions, tiredness or family visits (Price et al., 2018). Our study aimed to explore users’ motivations to report their pain levels without the additional trigger that is a periodic reminder or alarm. In this way, our study perhaps aligns more with adherence, a term that is related to compliance but in which patients have a more active role (Lutfey & Wishner, 1999). We found that most participants report pain when they feel it, and very few participants report pain that is decreasing or that they have no pain. Compliance with the self-report of pain is key for a successful monitoring, and our results help to understand that the self-report of pain on its own initiative presents very low values and is not correlated to activities or heart rate. We also note that our results—as the results of any similar study—suffer from compliance bias, since our data comes from the participants that did report their pain levels.

This study was conducted with 24 participants, which does not allow us to generalize its results to a broader population. However, relevant insights can be gathered from small populations, and sample size in Human-Computer Interaction (HCI) studies can vary widely according to method and study setting (Caine, 2016). Previous studies have found that in HCI studies, the number of participants can range from 1 to tens and hundreds of thousands (Caine, 2016; Koeman, 2018). Some researchers have proposed a minimum number of participants, e.g., seven (Turner, Lewis & Nielsen, 2006), 3–20 (Perfetti & Lori, 2015; Virzi, 1992), or depending on the complexity, e.g., five for simple studies (Nielsen, 2015). For qualitative studies, data should be collected until data saturation, i.e., “the point during qualitative data collection where no new relevant information emerges” (Caine, 2016). In our study, we faced limitations in the recruitment of older adult participants due to ethics committee regulations, inclusion criteria and the regulations of nursing homes. Even with these limitations, when we analyzed the qualitative data of the 24 participants, we reached a data saturation point with respect to the two research questions.

Regarding the usefulness of using a device such as TeRa, one notable outcome was that it enabled patients to improve the knowledge of their own pain, which is important to empower patients in their own pain management (Spyridonis et al., 2014), providing patients with a participatory role in their own health (Bird et al., 2016) and improving levels of interaction with healthcare professionals (Bird et al., 2016; Spyridonis et al., 2014). The fact that the older adults indicated that they would use the device again is the result of them finding it beneficial. This outcome is in line with previous research (Nedopil C., 2013).

The usability evaluation and user experience of TeRa were positive, which implies that using a tangible interface which involves simple interaction and provides haptic feedback may be appropriate for older adults to record their pain levels. These results confirm previous studies that have found tangible interfaces with buttons more applicable for data collection from older adults (Price et al., 2018). Moreover, in our study the majority of participants had little or no knowledge of such technology (20/24 people) and low levels of education. This factor is relevant, since it has been shown that people with higher levels of education are more likely to have a smart device (LaMonica et al., 2021); in our study seven people did not have a mobile phone and seven people only used it to call and receive calls. Therefore, it is necessary that monitoring technologies for older adults take into account this digital divide and facilitate adoption, e.g., through providing low-cost devices and training. Older adults living in nursing homes rated the usability of our prototype with a low score (79.32), so it is important to study other interaction approaches for this group of older adults, who have needs and limitations that may be somewhat different to more independent older adults.

The results may not be generalizable to other populations, who may find using a specialized device such as this one stigmatizing. We propose that this issue could be further studied by comparing acceptability of such a device with a smartwatch application, that is not as conspicuous, across populations of older adults. Several previous studies have used smartwatches and consumer wearables to track or detect health data, e.g., movement disorders (Varghese et al., 2020), inertial tracking of in-patients (Auepanwiriyakul et al., 2020), and in older populations, e.g., to detect walker use (Antos et al., 2019). Health trackers have been found to be acceptable for use in older adult populations (e.g., Batsis et al., 2016; Mercer et al., 2016). However, more work needs to be done regarding how older adults can use smartwatches not only to monitor their health information but also to self-report it.

Design insights

We derived some design insights from our results that could be used to design future prototypes of devices for older adults to self-report their pain, considering when older adults would prefer to report. This section provides a list of these design insights, along with a brief description of each.

1. Improving compliance. This study did not detect a need to trigger a reminder to report pain due to e.g., more strenuous activity or increased heart rate. Rather, each older adult has their own personal reasons to want to report pain or not (increased levels of pain, periodicity, finding their own correlations with their activities). Therefore, the strategies used by previous studies in which pain levels are requested periodically or at random intervals are probably best. These strategies could take into consideration the personal preferences of each user, e.g., for those who feel that a certain activity affects their pain, this activity could be detected and trigger a request for pain reporting, and for those who would like to understand trends in their pain, it should request pain reporting at the same time everyday.

2. Feedback and Visualization: Devices such as TeRa should provide a report, or visualization, of the information input by the users. This would allow older adults to understand their data and feel they have an active role with respect to their own health care. This information should also be able to be shared with health professionals or family members. This finding was also evidenced in LaMonica et al. (2021), where older adults mentioned several facilitators for the adoption of health information technologies, such as: personalization and the ability to access information.

3. Portability and weight: Pain self-report devices must be small and lights, so users can carry them anytime, anywhere; and also should be discreet, to allow users to report pain privately.

4. Clothing considerations: TeRa was placed in different locations during the experiment, with eight of the male participants choosing to insert it in their shirt pockets and ten of the female participants electing to carry it in a waist pouch. This implies that clothing plays a role in the placement of the device, since the clothes that were worn by men had pockets (the shirt), while the women’s clothing was more limited in this regard. This type of consideration regarding clothing should be taken into account when using tangible devices that need to be transported during the carrying out of daily tasks.

Limitations

Participants living in the nursing home have strict schedules regarding meals, medications, naps, baths, therapies, among others. Therefore, and due to these regulations, the time of use of TeRa was a brief intervention lasting three hours. In HCI studies, time is a factor that determines whether evaluations are made with users or not, due to the complexity they present (Goodman, Stolterman & Wakkary, 2011). In our case, this complexity increased, as we had to abide by nursing home guidelines. We acknowledge that use of such a device during prolonged times would provide additional insights about use and adoption. However, we consider that despite these drawbacks, due to the context of the target users, the results may be useful for future studies.

Although participants had low digital skills, we did not compare TeRa to another device (e.g., a smartphone), so it is not possible to conclude that tangible devices are better than other types of devices, even with participants with low digital skills. Another limitation is the number of participants in the study, since with 24 people it is not possible to generalize the results to a broader population. Likewise, a limitation of the study that must be taken into account when analyzing the results is the fact that when the participants did not report pain, it could be for a number of reasons, e.g., they did not have pain, they forgot to report the pain, or they were in too much pain to report it.

Conclusions

Participants self-reported their pain when they were feeling pain, and very few people reported that their pain was decreasing or that they had no pain. The TeRa device was found to have good usability and a positive user experience. Older adults, even those with low digital skills, were able to use the device successfully. The location in which the tangible device is transported may vary depending on the user.

In terms of future work, the next step in our research is to incorporate these design insights into TeRa, to create a version of the device that is more appropriate for older users who want to register and report pain levels. Then, we plan to validate the new version of TeRa by comparing it with other types of non-tangible interfaces, in order to identify if there are significant differences. The idea is to study the adoption of the device over time through qualitative and quantitative analysis, i.e., monitor compliance with pain reporting as well as uncover participants’ reasons for using or not using the device as expected. We also plan to study how to leverage the device to improve the relationship between users and healthcare professionals. Finally, it would be valuable to allow users to explicitly include with their report, what triggered them to report their pain. However, future studies should propose how to do so in a way that is unobtrusive and does not interrupt users or inconvenience them.

Supplemental Information

Supplemental Information 1 Data from the participants in the questionnaires, the interviews, and in the observations

Click here for additional data file.

We would like to thank the older adults who participated in this research and the nursing homes that opened their doors to us.

Additional Information and Declarations

Competing Interests

Author Contributions

Human Ethics

Data Availability

The authors declare there are no competing interests.

Iyubanit Rodríguez conceived and designed the experiments, performed the experiments, analyzed the data, prepared figures and/or tables, authored or reviewed drafts of the article, ethical and looking for participants issues, and approved the final draft.

Gabriela Cajamarca conceived and designed the experiments, authored or reviewed drafts of the article, collect data, and approved the final draft.

Valeria Herskovic conceived and designed the experiments, performed the experiments, authored or reviewed drafts of the article, administration issues, and approved the final draft.

The following information was supplied relating to ethical approvals (i.e., approving body and any reference numbers):

The research protocol was approved by the Ethics committee of the Medical School of the Pontificia Universidad Católica de Chile(Ethical Ref. 15-339).

The following information was supplied regarding data availability:

The raw data is available in the Supplemental File.

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
