# Peer review of "When does self-report of pain occur?: A study of older adults"

_PeerJ, doi:10.7717/peerj.13716_

## Round 0.1 · original submission · Major Revisions

Two detailed reports have been received. Both reviewers find the paper is valuable but a major revision is needed before further processing. Please also provide a detailed response letter. Thanks.

Reviewer 1 ·

Basic reporting

Thank you for the opportunity to review the manuscript entitled “When does self-report of pain occur?: A study of older adults”. The manuscript deals with a topical concern, i.e., self-reporting technologies among older adults. The authors test a simple tangible interface to self-report pain levels among 24 older adults.

Abstract: The abstract is clear and comprehensive. The authors introduce the background, describe the methodology, clarify the aims and scope of the research and highlight the main results and outcomes, as well as some limitations (e.g., limited time frame of the experiment).

Introduction: The section “Introduction” is clear. The authors state the general problem and introduce some relevant aspects of the issue. Further, the authors propose a literature review of the topic, introducing in a clear and comprehensive way the difference between tangible user interfaces and digital technologies. However, I suggest the authors including more recent studies, since several of them date back to the early 2000s or even before. For instance, is there the chance to quote more recent articles related to: “this process can be affected by memory bias” (line 30) or “pain can vary on a daily basis” (line 33)?
Although significant, I would avoid saying: “The short duration of the study due to nursing home regulations - means that we were not able to study long term adoption or patterns” (lines 80-81). I would include them in the sub-section “Limitations” at the end of the manuscript.

Experimental design

Materials and methods: The section “Materials and methods” describes the experiment and the prototype. The authors include references to justify the adoption of some features (e.g., buttons, sound effects). Is there the chance to define what “LilyPad Arduino” is?

As regards the study context, the authors describe the sampling techniques: first, the snowballing sampling technique; second, the list of people in the nursing home. However, the authors should better define how the sampling technique has been implemented, and how the nursing home has been selected. Further, they refer to “a list of people who met the inclusion criteria”. Could the authors define these “inclusion criteria”?

Further, as concerns the “oral interview”, how was it structured? Which questions have been asked to the participants? Please clarify. If this oral interview is the same described at lines 194-199, please try to make it clearer.

In the field of “description of study participants”, I would move this paragraph to the beginning or the section “Results”.

The description of the experiment is clear.

As regards the description of the “quantitative analysis” (lines 201-205), the authors should refer to successful studies applying the same methodology. It helps in strengthening the research.

Validity of the findings

Results: The research questions presented in the section “Introduction” are not the same presented in the section “Results”. Please, use the same research questions, in order not to create confusion among readers.

Results are presented in a comprehensive and clear manner. Descriptive data are supplemented by figures and tables. Further, I agree with the inclusion of participants quotations.
Discussion: The section “Discussion” and the sub-section “Design Insights” are clear. The authors present novel issues and refer to several previous studies, highlighting the transversality and the utility of the present research. The authors refer to the above-mentioned research questions and follow a coherent approach along the entire manuscript.

Conclusions: Conclusions are adequate and tie with the entire research.

Additional comments

The authors should respect the “guide for authors” as regards in-text references. References should be included within brackets in the manuscript.

·

Basic reporting

Self-reporting of pain with the use of technology is a relevant and promising area of work for improving the quality of life of older adults. This is important work, however, there are few limitations, which I would like to highlight and would want to see how authors are addressing them in their research.
In text referencing style is not appropriate throughout. At least, I have not seen that style of referencing. It is certainly limiting my ability to read.
English should be proofread throughout. There were grammatical errors.
Regarding introduction: Focus on older population for pain self-reporting is relevant and important. Authors have mentioned key challenges for older adults or elderly population in self-reporting pain using digital technology. However, cognitive impairment and dexterity are other two important aspects, which I would suggest authors to consider or include. Also, there is a general critique around single rating scales for capturing complexity in pain. Also, measurement errors and validation issues have been reported previously. Given that, please justify, why it was essential to use single rating scale, and how it is more beneficial than other pain assessment tools?

Experimental design

Experiment
“Each experiment had a duration of between 230 and 245 minutes per participant”, reading it feels like it was done in one session. Please rephrase and clarify. If it is a longitudinal study?
Factors that trigger people’s pain reporting behaviour is very interesting aspect of pain self-reporting, however, that is limited only to emotional state, patient activity and heart rate. Could participant submit any pain diary to mention those aspects, which above stated pain triggers could not capture? Also, patient activity and heart rate were measured in real-time using wristband, how did you capture emotional state in real-time?

Validity of the findings

Results
Not clear, how data on emotional state was collected and analysed.
Participant could report their pain, whenever they had a pain sensation. It means, no pain intensity score will be assumed as such, without considering a design flaw. For example, a participant may forget, or is in too much pain that limited his ability to report pain. If this is not considered then I would say caution should be considered in interpreting the results.
I can understand that there is a methodological challenge of summarising multiple data points across the population, but I would like to see how this method of data collection can be used for population health research.

Additional comments

My biggest concern is use of single pain rating scale. A clear justification for it might be helpful in addressing my concern

---

## Round 0.2 · accepted · Accept

The paper can be accepted. Congratulations.

Reviewer 1 ·

Basic reporting

Thank you for the opportunity to read and review the revised version of the manuscript entitled "When does self-report of pain occur?: A study of older adults". I have carefully read the responses of the authors to my comments/suggestions, as well as the entire revised manuscript, which seems to me suitable for publication in the Journal. The authors have substantially revised their research and added several novel (and more recent) details, reaching a good quality standards and an adequate scientific soundness. I am convinced of the importance of the research.

Experimental design

N/A

Validity of the findings

N/A

Additional comments

N/A

·

Basic reporting

Authors have considered all my previously made suggestions, and responded each of them appropriately. I have no further comment or concern

Experimental design

Authors have considered all my previously made suggestions, and responded each of them appropriately. I have no further comment or concern

Validity of the findings

Authors have considered all my previously made suggestions, and responded each of them appropriately. I have no further comment or concern

Additional comments

Authors have considered all my previously made suggestions, and responded each of them appropriately. I have no further comment or concern